# Genetic Analyses of Seed Longevity in *Capsicum annuum* L. in Cold Storage Conditions

**DOI:** 10.3390/plants12061321

**Published:** 2023-03-14

**Authors:** Mian Abdur Rehman Arif, Pasquale Tripodi, Muhammad Qandeel Waheed, Irfan Afzal, Sibylle Pistrick, Gudrun Schütze, Andreas Börner

**Affiliations:** 1Nuclear Institute for Agriculture and Biology, Faisalabad 38000, Pakistan; 2Research Centre for Vegetable and Ornamental Crops, Council for Agricultural Research and Economics (CREA), 84098 Pontecagnano Faiano, Italy; 3Seed Physiology Lab, Department of Agronomy, University of Agriculture, Faisalabad 38000, Pakistan; 4Leibniz Institute of Plant Genetics and Crop Plant Research, Corrensstr. 3, 06466 Seeland, Germany

**Keywords:** genetics, candidate genes, GWAS, *Capsicum*, genebanks, seed longevity, cold storage

## Abstract

Seed longevity is the most important trait in the genebank management system. No seed can remain infinitely viable. There are 1241 accessions of *Capsicum annuum* L. available at the German Federal ex situ genebank at IPK Gatersleben. *C. annuum* (*Capsicum*) is the most economically important species of the genus *Capsicum*. So far, there is no report that has addressed the genetic basis of seed longevity in *Capsicum.* Here, we convened a total of 1152 *Capsicum* accessions that were deposited in Gatersleben over forty years (from 1976 to 2017) and assessed their longevity by analyzing the standard germination percentage after 5–40 years of storage at −15/−18 °C. These data were used to determine the genetic causes of seed longevity, along with 23,462 single nucleotide polymorphism (SNP) markers covering all of the 12 *Capsicum* chromosomes. Using the association-mapping approach, we identified a total of 224 marker trait associations (MTAs) (34, 25, 31, 35, 39, 7, 21 and 32 MTAs after 5-, 10-, 15-, 20-, 25-, 30-, 35- and 40-year storage intervals) on all the *Capsicum* chromosomes. Several candidate genes were identified using the blast analysis of SNPs, and these candidate genes are discussed.

## 1. Introduction

Seeds are considered the building blocks of genebanks, which came in to being to preserve plant genetic resources and avoid the risk of extinction and of genetic erosion [1]. Seed storage in the genebanks also ensures the preservation of allelic (or genic) combinations in germplasm collections [2], thus serving as the raw material to breed new cultivars [3]. The success of plant genetic resources stored in genebanks was recently realized for wheat [4] when several genebank lines of wheat from Mexican genebank were crossed with several elite cultivars developed at CIMMYT [4] to produce a large number of different pre-breeding germplasm sets and were distributed to resource-poor countries including India and Pakistan. The resultant germplasm revealed considerable success in providing favorable alleles for disease resistance [5], salinity tolerance [6], nitrogen-use efficiency [7], nematode resistance [8], Karnal bunt resistance [9] and drought tolerance [10]. More recent evaluations of Mexican wheat landraces coupled with genetic mapping have also revealed their latent potential toward food security in the upcoming decades [11].

Global food supply relies on the availability of viable seeds [12]. To maintain their germinability, the genetic resources stored in the form of seeds need to be regularly evaluated [13]. A drop in their germination beyond a certain threshold indicates that regeneration is required [14]. This renders “seed longevity” the single most important trait in the genebank management system [15]. Seed longevity refers to the time period during which a seed remains viable and capable of producing healthy seedlings [16,17]. Research on seed longevity is of extreme significance to genebank management [12]. No seed can remain infinitely viable. Conditions during seed production, crop harvesting, post-harvest conditions and, later, the storage conditions determine seed viability [18]. Seed viability, however, is also variable among species and even between varieties, indicating that the genetic component also plays an important role in determining seed longevity [3].

Vegetable seeds constitute ~19,000 accessions of the IPK germplasm collection [19]. According to the genebank information system (GBIS) of the IPK (https://gbis.ipk-gatersleben.de/gbis2i/faces/index.jsf, accessed on 20 October 2022), there were 1241 accessions of *Capsicum annuum* L. available. *C. annuum* is the most economically important species of the genus *Capsicum* [20]. A recent report has shed considerable light on its evolution and trade history in addition to mapping genes related to its plant architecture, fruit quality and flower-related traits [21]. Genetic analyses of seed longevity, however, in *Capsicum* are non-existent. Recently, however, the molecular mechanisms involved in seed longevity in different *Capsicum* species and varieties were illustrated. Less domesticated species (*C. chinense* and *C. frutescens*) exhibited higher germination rates and longevity after AA. Differential gene expression analyses exhibited that *aspartic protease in guard cell 1 (ASPG1)* and *homeobox protein 25 (HB25)* expression were higher in long-lived accessions. In addition, a positive correlation between the amount of lignin and seed viability was demonstrated [20].

Among the two most common techniques to investigate the genetic components of a trait, association mapping (AM) is advantageous to conventional linkage mapping technique because AM does not require the generation of a defined “population”. AM utilizes unrelated accessions or collections of germplasm [3]. AM has been performed on a range of crop species with respect to seed longevity analysis, including *Arabidopsis* [22], rice [23], barley [24] and wheat [14]. Most of the studies that analyzed seed longevity involved the lab-based AA or CD tests, the results of which in major crop plants such as wheat have remained under debate in comparison to long-term cold-storage aging [14]. Moreover, seed longevity assessment under lab conditions is costly due to the growing of a plant for one complete season, harvesting it and subsequently storing it, followed by experimental protocols that involve seed aging at high temperatures and high relative humidity [15]. Here, we report on the molecular genetic analyses of seed longevity in *Capsicum annuum* L. by using the genebank germination data generated over a period of 40 years and employing the AM protocol.

## 2. Results

### 2.1. Standard Germination after Various Storage Periods

The standard germination percentage after various storage periods varied considerably. For example, the germination % after 1–5 years of storage was considerably high with a mean value (±standard deviation) of 86.08 ± 14.82%, whereas the germination % after 36–40 years of storage was 64.49 ± 23.87% (Figure 1). The germination % after 6–10 years dropped sharply to 71.63 ± 26.84%. However, mean germination % after 11–15 and 16–20 years of storage remained 80.33 ± 18.15% and 80.09 ± 21.14%, respectively. Likewise, germination % during the periods of 21–25 and 26–30 years of storage dropped minimally, with mean values of 77.64 ± 18.30% and 77.73 ± 15.92%, respectively. Afterward, there was some decline in survival after 31–35 years of storage when germination% was 70.06 ± 18.28%.

### 2.2. Genome-Wide Association (GWA) Mapping

Association mapping was carried out separately for each storage interval. We identified a total of 34 significant maker trait associations (MTAs) (including 10 highly significant MTAs) for the longevity of *Capsicum* seeds stored from 1 to 5 years (Appendix A, Figure 2). These MTAs were located on chromosomes 1 (6 MTAs), 2 (2 MTAs), 4 (2 MTAs), 5 (3 MTAs), 6 (2 MTAs), 7 (4 MTAs), 9 (4 MTAs), 10 (6 MTAs), 11 (3 MTAs) and 12 (2 MTAs), which explained 1.6–7.6% phenotypic variance. For the storage period of 6–10 years, 25 significant MTAs were detected on chromosomes 1 (1 MTA), 2 (3 MTAs), 3 (2 MTAs), 4 (1 MTA), 5 (2 MTAs), 6 (2 MTAs), 7 (3 MTAs), 8 (1 MTA), 9 (8 MTAs), 10 (1 MTA) and 12 (1 MTA). These MTAs explained 3.3 to 6.6% phenotypic variance. Likewise, 31 significant MTAs (including 1 highly significant MTA) were detected for germination after the storage period of 11–15 years, and these MTAs were located on chromosomes 1 (5 MTAs), 2 (1 MTA), 3 (2 MTAs), 4 (4 MTAs), 5 (1 MTA), 6 (1 MTA), 7 (1 MTA), 8 (2 MTAs), 9 (2 MTAs), 10 (7 MTAs), 11 (4 MTAs) and 12 (2 MTAs). These MTAs were responsible for 3.0 to 8.0% phenotypic variation.

For the longevity after 16–20 years of storage, another 35 significant MTAs (including 2 highly significant MTAs) were found. These were exhibited on chromosomes 1 (2 MTAs), 2 (3 MTA), 3 (2 MTAs), 4 (5 MTAs), 5 (2 MTAs), 6 (5 MTAs), 7 (3 MTAs), 8 (1 MTA), 9 (7 MTAs), 10 (2 MTAs), 11 (1 MTA) and 12 (1 MTA), which explained between 2.3 and 4.5% differences in longevity. The highest number of MTAs (39 significant MTAs including 5 highly significant MTAs) for longevity were detected after 21–25 years of storage. Here, the chromosomes involved were 1 (2 MTAs), 2 (4 MTA), 3 (1 MTA), 4 (3 MTAs), 5 (4 MTAs), 6 (2 MTAs), 7 (6 MTAs), 9 (3 MTAs), 10 (5 MTAs), 11 (8 MTAs) and 12 (1 MTA), and the variation explained was 9.7–23.9%. On the contrary, the least number of MTAs (7 significant MTAs including 2 highly significant MTAs) were detected for the storage period of 26–30 years. These were located on chromosomes 2 (3 MTAs), 4 (1 MTA), 6 (1 MTA), 7 (1 MTA) and 10 (1 MTA). The phenotypic variance explained, however, was 8.0–14.3%.

In the case of storage for 31–35 years, 8 different chromosomes [chromosome 2 (2 MTAs), 3 (1 MTA), 4 (2 MTAs), 6 (2 MTAs), 7 (2 MTAs), 9 (9 MTAs), 10 (2 MTAs) and 11 (1 MTA)] carried 21 significant MTAs, and the variance explained was 1.8–3.9%. Finally, another 32 MTAs (including 2 highly significant MTAs) were discovered for longevity after storage for 36–40 years, which involved all the *Capsicum* chromosomes except chromosome 8. These MTAs were located on chromosomes 1 (3 MTAs), 2 (1 MTA), 3 (2 MTAs), 4 (5 MTAs), 5 (1 MTA), 6 (4 MTAs), 7 (1 MTA), 9 (5 MTAs), 10 (3 MTAs), 11 (2 MTAs) and 12 (6 MTAs), which were responsible for a 8.6–30.2% difference in longevity. Thus, 224 MTAs (including 22 highly significant MTAs) collectively were detected on all the *Capsicum* chromosomes for longevity after various years of storage.

## 3. Discussion

### 3.1. Variation in Germination over Various Periods of Storage

Seed deterioration depends on many factors encompassing the environmental and genetic components [14]. This was exhibited in wheat when seeds from the multiplication year of 1974 were stored and tested 34 years later, and germination % varied from 0 to 100%. Likewise, a huge variation in seed survival was witnessed albeit after storage below freezing temperatures (−15/−18 °C) in *Capsicum*. A direct comparison among the results of different intervals is not possible because of the involvement of dissimilar accessions tested during each interval. However, some accessions were found to be common across different intervals. For example, there were 171, 261, 419, 163, 108, 318 and 54 accessions from 1- to 5-year intervals that were common in the 6–10-year, 11–15-year, 16–20-year, 21–25-year, 26–30-year, 31–35-year and 36–40-year intervals, respectively. ANOVA results indicated that germination % among common accessions was significantly different in all such cases (data not shown). No reports exist in which such comparisons were made in any crop. It is, however, known that different factors are involved in different aging procedures such as accelerated aging (AA) and controlled deterioration (CD) methods [3]. A comparison between natural aging (seeds stored at 0 ± 1 °C at constant moisture contents of 8 ± 2%) and the deterioration of fresh seeds after AA and CD yielded different results in wheat [14].

### 3.2. GWA Analyses and Candidate Genes

The seed lots were handled in the same way (from seed sowing to harvest and post-harvest treatments) and were maintained at the IPK genebank since then. The differential behavior in the investigated material was thought to be due to differences in the genetic build of these accessions. Our analyses identified a total of 224 MTAs for 8 different years of storage periods in which chromosome 9 carried the highest number of MTAs (38 MTAs), followed by chromosome 10 (27 MTAs), followed by chromosome 4 (23 MTAs) (Appendix A). Chromosome 7 carried 21 MTAs, whereas chromosomes 1, 6 and 11 carried 19 MTAs each. On the other hand, chromosome 2 carried 16 MTAs, and 13 MTAs were located on each of chromosomes 3 and 5. Finally, 12 and 4 MTAs were located on chromosomes 12 and 8, respectively. Thus, this is the very first report of the GWA of seed longevity in *Capsicum*; no comparison with previous studies can be made. Blast analyses of the reported SNPs identified several candidate genes for longevity (Appendix A). Of the 220 associated SNPs, 167 SNPs successfully provided hits with certain candidate genes. These 167 SNPs could further be divided into 5 groups based on the function they perform (Figure 3, Appendix A). The first group included ten genes which were involved mainly in growth- and development-related processes. The other group constituted 72 genes which were mainly enzymes that were either specifically produced under (both biotic and abiotic) stress or produced under normal conditions. The third group constituted 10 genes that were mainly transcription factors. The fourth group included 18 genes that were mainly transporter genes, whereas the fifth group constituted 48 genes that were mainly uncharacterized and/or hypothetical proteins.

In the following, we provide some details (chromosome by chromosome) for the candidate genes (Appendix A) that are linked with SNPs that explain >4% phenotypic variation or are reported to be associated with longevity in other crops.

On chromosome 1, we identified subtilisin-like protease 4 (associated with SNP S1_7853500), 3-oxoacyl-[acyl-carrier-protein] (ACPs) synthase I (associated with SNPs S1_1430591 and S1_1435099), phosphatidylinositol 4-kinase gamma 2 (associated with SNP S1_1921287) and B3-domain-containing transcription factor NGA1 isoform X2 (associated with SNP S1_133318). Subtilisin-like proteases (subtilases) are serine proteases that play specific roles in plant development and signaling cascades. Several subtilases are specifically induced following pathogen infection or under stress [25]. They are also identified as the S-nitrosylation target in potato *S*-nitrosylation candidates in the potato–*Phytophthora infestans* system [26]. On the other hand, ACPs are a central cofactor for de novo fatty acid synthesis, acyl chain modification and chain-length termination during lipid biosynthesis in living organisms. Different ACP isoforms have been found to be responsible for the biosynthesis of fatty acids and lipids for specific purposes in plants [27]. In addition, ACP has also been identified as a candidate gene for resistance against different insects [thrips, orange (OWBM) and yellow (YWBM) wheat blossom midges] in wheat [28]. Similarly, phosphatidylinositol 4-kinase gamma 2 is known to play a role in the phosphorylation of phosphatidylinositol (PI) to PI 4-phosphate, which is one of the key reactions in the production of phosphoinositides, which are lipid regulators of several cellular functions [29]. NGA transcription factors are involved mainly in developing pistils; they are also involved in regulating the shape and size of lateral organs such as leaves and petals and the regulation of seed size [30].

On chromosome 2, the candidate genes causing significant variation toward seed survival include putative *leucine-rich repeat receptor-like protein kinase* (associated with SNP *S2_59895267*), putative *uroporphyrinogen decarboxylase, chloroplastic-like* (associated with SNP *S2_117264835*), putative *aspartic proteinase nepenthesin-2-like* (associated with SNP *S2_157010230*) and *THO complex subunit 2-like* (associated with SNP s*S2_165845447* and *S2_165845425*). *Aspartic proteinase nepenthesin-2-like* was reported during the periods of 11–15 years and 20–25 years of storage. The nepenthesin aspartic proteases, which are produced by specialized cells in the lower part of the pitchers, are aimed primarily at the digestion of prey trapped by the plant [31]. *Aspartic protease in guard cell 1* has recently been reported as a candidate gene for longevity in *Capsicum* [20]. *Aspartic proteases* mobilize seed-storage proteins and play a crucial role in the germination process and seed longevity [32]. Likewise, the THO complex that is encoded by *THO complex subunit 2-like* is a key component in the co-transcriptional formation of messenger ribonucleo-particles that are competent to be exported from the nucleus (unknown precise function). The THO complex is also involved in mRNA processing and its transport from the nucleus. It also plays a role in small interfering RNA-dependent processes in plants [33,34]. The importance of the *THO complex subunit 2-like* is also evident from the fact that it was identified as a candidate gene for longevity during the periods of 1–5 years and 21–25 years of storage and on multiple chromosomes (chromosome 2 and 11).

The candidate genes for longevity on chromosome 3 include mitochondrial NADH dehydrogenase (ubiquinone) flavoprotein 2 (associated with SNP S3_145194851), chalcone synthase 1B (associated with SNP S3_7697933) and ethylene-responsive transcription factor 4 (associated with SNP S3_12004386). Both NADH dehydrogenase and ethylene-responsive transcription factor have previously been reported as candidate genes for seed dormancy/pre-harvest sprouting (PHS) (in wheat) [35] and longevity (in wheat and barley) [13,36], respectively. Chalcone synthase (CHS) is a crucial rate-limiting enzyme in the flavonoid biosynthetic pathway that catalyzes the condensation of malonyl-CoA and ρ-coumaroyl-CoA to produce naringenin chalcone, which serves as the precursor of a variety of flavonoid derivatives. These flavonoids are involved in the response to and protection of plants from abiotic and biotic stress, including ultraviolet radiation, temperature, humidity and pathogenic attack [37,38]

The most important candidate genes on chromosome 4 include *early nodulin-93 isoform X2 *(associated with SNP *S4_15009085*), *transcription factor TGA7* (associated with SNP *S4_209765879*), *3-ketoacyl-CoA synthase 19* (associated with SNP *S4_124426*) and *vacuolar protein-sorting-associated protein 8 homolog* (associated with SNPs *S4_28351186* and *S4_28351189*). *Early nodulin* has been identified as a candidate gene for longevity, dormancy and PHS in wheat [3,35]. The *3-ketoacyl-CoA synthase* is involved in lateral organ development and cuticular wax synthesis in *Medicago truncatula* [39]. The *TGA* family of transcription factors plays important roles in the systemic acquired resistance (SAR) in plants. However, despite its important roles in plant immunity, the molecular mechanism for the DNA binding of *TGA7* remains unclear [40]. Vacuolar protein-sorting-associated proteins (Vps) are part of the Endosomal Sorting Complex Required for Transport (ESCRT), which performs the topologically unique membrane bending and scission reaction away from the cytoplasm [41]. 

Scarecrow-like protein 30 (associated with SNP *S5_226924789*), ankyrin repeat-containing protein 2A and ITN1 (associated with SNPs *S5_238134029* and *S5_237978262*, respectively), protein ACCELERATED CELL DEATH 6 (ACD6) (associated with SNP *S5_237978236*), GRF1-interacting factor 1 (associated with SNP S5_26823246), histone acetyltransferase HAC1-like (associated with SNP *S5_14262255*) and putative LRR receptor-like serine/threonine-protein kinase-like (associated with SNP S5_24462479) were among the candidate genes on chromosome 5. Scarecrow-like protein is a transcription factor belonging to the GRAS family. It regulates root growth and the cell cycle and also mediates resistance to environmental stresses [42]. Recently, 85 ankyrin repeat-containing protein (ANK) genes in *C. annuum* were identified. Our ANK loci on chromosome 5 (SNPs: *S5_237978262* and *S5_238134029*) could correspond to any of the CaANK35-CaANK51 genes mapped at the distal end of chromosome 5 on the *C. annuum* L. genome [43]. ANKs have also been identified as candidate genes against insect (OWBM and YWBM) resistance in wheat [28]. ACD6 is a multipass membrane protein with an ankyrin domain that acts in a positive feedback loop with the defense signal salicylic acid (SA) [44]. GRFs are a class of plant-specific proteins involved in the regulation of stem and leaf development that act mainly as positive regulators of cell proliferation [45]. Histone acetyltransferase HAC1-like encode for histone acetyltransferases that play a crucial role in the control of cell fate and influence cell cycle progression, plant responses to environmental conditions, and gene interactions [46,47]. LRR receptor-like serine/threonine-protein kinase-like was also reported for the longevity-associated SNP (S2_59895267) on chromosome 2 and functions in protein phosphorylation and the transmembrane receptor protein tyrosine kinase signaling pathway. It is an integral component of the plasma membrane, where it functions as an ATP-binding site and is expressed in the flowering stage and plant embryo stage in flowers or seeds [48]. It has also been detected as a candidate for seed longevity in wheat [13].

Among the candidate genes on chromosome 6, the most important were *ribose-phosphate pyrophosphokinase 1-like* (associated with SNP *S6_196787696*), *11S globulin seed storage protein* (associated with SNP *S6_1150637*), *beta-galactosidase* (associated with SNP *S6_213563914*) and putative *ATP synthase subunit O, mitochondrial-like* (associated with SNP *S6_182246997*). *ATP synthase subunit O, mitochondrial-like* was also associated with SNPs on chromosomes 7 (SNP: *S7_157986416*) and 9 (SNP: *S9_35876497*)*. Ribose-phosphate pyrophosphokinase,* which is also known as *phosphoribosyldiphosphate synthetase* (PRPP), catalyzes the biosynthesis of PRPP. PRPP is a precursor for the synthesis of pyrimidine, purine, pyridine nucleotides, tryptophan and histidine [49]. Plant seed storage proteins function as the major nitrogen source for the developing plant. The 11S-type globulins are non-glycosylated proteins which form hexameric structures. They are the proteins required for the development or growth of seeds [50]. *Beta-galactosidase* is considered to be an important regulator involved in fruit ripening in *Capsicum* [51]. The reduced form of the *mitochondrial ATP synthase* holoenzyme leads to wide-ranging defects in energy-demanding cellular processes. Hence, it is required to protect plants from various stresses such as heat [52].

Important genes on chromosome 7 include pentatricopeptide repeat-containing protein-mitochondrial-like (associated with SNP S7_42146593), NADP-dependent glyceraldehyde-3-phosphate dehydrogenase (associated with SNP S7_8830422), COP9 signalosome complex (CSN) subunit 8-like (associated with SNP S7_244455030), 1-acyl-sn-glycerol-3-phosphate acyltransferase (associated with SNP S7_245349068) and protein phosphatase 2C 27 (associated with SNP S7_13703435). Here, pentatricopeptide repeat-containing protein was associated with longevity after 11–15 years, 21–25 years and 26–30 years of storage. Pentatricopeptide repeat-containing protein [members of the pentatricopeptide repeat (PPR) protein] family are sequence-specific RNA-binding proteins that play crucial roles in organelle RNA metabolism [53]. In addition, PPR is also involved in YBWM resistance in wheat [28]. PPRs have also been identified as candidate genes that are involved in seed vigor under low-temperature conditions in rapeseed [54]. Glyceraldehyde-3-phosphate dehydrogenase is used in a variant of glycolysis that conserves energy as NADPH rather than as ATP [55]. CSN is an evolutionarily conserved multiprotein complex that regulates many aspects of plant development [56]. In addition, the glycerol-3-phosphate acyltransferase gene plays a pivotal role in cold resistance in a variety of plant species [57], whereas a Type 2C protein phosphatase, CaADIP1 (Capsicum annuum ABA and Drought-Induced Protein phosphatase 1), is known to be expressed on leaves on treatment with ABA, drought and NaCl treatments [58].

Putative *cysteine synthase* (associated with SNP *S8_140480324*) and *syntaxin-32-like* (associated with SNP *S8_119314147*) were the candidate genes on chromosome 8. The former is an enzyme responsible for the formation of cysteine from *O*-acetyl-serine and hydrogen sulfide with the concomitant release of acetic acid [59], and the latter is reported to be involved in host defense responses against pathogen attack [60].

The most important candidate genes on chromosome 9 include salicylate O-methyltransferase-like (SAMT) (associated with SNPs S9_91765981 and S9_91766032), solute carrier family 35 member F1-like (associated with SNP S9_649935), cell division cycle protein 48-like protein (associated with SNP S9_5554728), superoxide dismutase (associated with SNP S9_93034919) and eukaryotic translation initiation factor isoform 4G-1-like isoform 1 (associated with SNP S9_270289444). SAMT regulates the SA signaling pathway and catalyzes the methylation of SA with *S*-adenosyl-l-methionine as the methyl donor to form methyl salicylate. SAMT appears to play an important role in plant response to drought stress by modulating the SA-signaling pathway [61]. Solute carrier family 35 member F1-like is akin to osmotin-like protein (OSML81). OSMLs belong to the thumatin-like protein family and are known to play a role in seed longevity in wheat and barley [35,36]. Cell division cycle proteins are known to be involved in cell division, growth processes and seed longevity [13]. Likewise, superoxide dismutase and eukaryotic translation initiation factor isoform 4G-1-like isoform 1 are also reported to be candidate genes for seed longevity [14] and PHS [35] in wheat.

On chromosome 10, the most important candidate genes include putative *histone H3.3-like* (associated with SNP *S10_374592* and multiple seed storage durations) and *high mobility group B protein 6* (associated with SNP *S10_16302829*). *Histone H3.3-like* is a candidate gene for longevity in wheat [3], whereas the gene encoding the “*high-mobility group B protein 6*” is a *WRKY transcription factor* involved in the nucleosome/chromatin assembly [62].

Chromosome 11 carries longevity genes such as putative flavin-containing monooxygenase 1-like (FMO) (associated with SNP S11_152967951), protein YAE1 isoform X3 (associated with SNP S11_257151051) and putative F-box protein-like (associated with SNP S11_257213134). FMOs are oxidoreductases and possess remarkable diversity and functionality in the oxygenation reactions, which are crucial steps within hormone metabolism, pathogen resistance, signaling and chemical defense [63]. YAE1 proteins are essential for growth under aerobic conditions and may provide protection from damage due to reactive oxygen species [64]. CaF-box is known to be expressed mainly in stems and seeds, and the transcript is markedly up-regulated in response to cold stress, ABA and SA) treatment, and down-regulated under osmotic and heavy metal stress [65]. However, its role in seed longevity is unknown. Finally, on chromosome 12, the most important candidate gene was tonoplast dicarboxylate transporter (associated with SNPs S12_2792525, S12_2792536 and S12_2792561), and it is known to play an important role in malate and citrate transport (organic acid metabolism) [66].

## 4. Materials and Methods

### 4.1. Materials

A total of 1152 *Capsicum* accessions was convened in this investigation. These were deposited in the IPK-Gatersleben over a number of years (from 1976 to 2017) (Appendix A) and kept in glass containers (Figure 4). A large proportion of them, however, were deposited during the years 1976 (165 accessions), 1977 (130 accessions) and 1978 (115 accessions). Of these accessions, 1137 were of *Capsicum annuum* L., 14 were *C. annuum* var. *glabriusculum,* and 1 was *C. frutescens.* These accessions were stored at below-freezing temperatures (1976–2010 −15°C; since 2011 −18 °C). Their details can be accessed through the Gatersleben genebank information system by providing the ID (identity number) of the accessions.

### 4.2. Standard Germination Tests

We performed 3103 germination assays on 1152 *Capsicum* accessions from 1990 to 2022. Not all accessions were assessed after each year. For easy understanding, we divided the germination assays into 8 intervals: (1) germination assays performed between 1 and 5 years of storage (812 tests), (2) germination assays performed between 6 and 10 years of storage (320 tests), (3) germinations assays performed between 11 and 15 years of storage (389 tests), (4) germinations assays performed between 16 and 20 years of storage (506 tests), (5) germination assays performed between 21and 25 years of storage (163 tests), (6) germination assays performed between 26 and 30 years of storage (169 tests), (7) germination assays performed between 31 and 35 years of storage (604 tests) and (8) germinations assays performed between 36 and 40 years of storage (140 tests) (Appendix A). Stored seeds in genebanks over many years are precious materials, and hence, only a limited quantity could be made available for research. Because of that, one single replicate of 50 seeds of each accession was retrieved from the glass containers and germinated on round filter paper with glass covers on Jacobsen Apparatus at 25 ± 2 °C and 23 ± 2 °C during the day and night, respectively. The germination percentages were recorded on the eighth day according to International Seed Testing Association (ISTA) protocols.

### 4.3. Genotyping

For the purpose of genotyping, 100 mg fresh leaf tissue that was collected from individual plants upon germination was used for DNA extraction. DNeasy Plant Mini Kit (QIAGEN, Düsseldorf, Germany) or the Sbeadex maxi plant kit (LGC Genomics, London, UK) was used to extract the highest-quality DNA, the quantity and quality parameters of which were determined using both spectrometry (ND-1000; NanoDrop, ThermoScientific, Wlatham, MA, USA) and fluorometry (Qubit 2.0 Fluorometer, Invitrogen, Carlsbad, CA, USA) methods. Samples with 260/280 and 230/260 ratios ranging between 1.8 to 2.2 and 1.8 to 2.0, respectively, and with a less-than-twofold deviation between fluorimetric and spectrophotometric readings were subjected to genotyping-by-sequencing (GBS). Genotyping was carried out using an Illumina HiSeq2500 platform generating 1 × 107-bp single-end reads version 3 chemistry (Illumina, San Diego, CA, USA), which resulted in the generation of 23,462 single nucleotide polymorphism (SNP) markers covering all the 12 *Capsicum* chromosomes. Other relevant details including the details of bioinformatics techniques and tools are available from Tripodi et al. [21].

### 4.4. Genome-Wide Association (GWA) Analyses

We performed the GWA analyses by utilizing the data of 23,462 high-quality SNP markers [21] and the data of standard germination tests that were obtained as mentioned above. The MLM (mixed linear model) option implemented in the *TASSEL v5.2.43* [67] software was used. A pre-requisite of this model is the provision of the population structure (Q-matrix) or principal component analysis (PCA) matrix and a kinship (K-matrix) matrix. These matrices are used as covariates in the MLM model to avoid false positives during analyses. The PCA matrix and the K-matrix could be generated through *TASSEL v5.2.43*. We ran each of the analyses using the PC = 3, PC = 4 and PC = 5 options for correct estimates. We found that 3, 4 or 5 PCs yielded 90–95% similar marker trait association with slight variation in the *p*-values of the associated SNPs. Thus, we kept PC = 5 for the final analysis. We claimed the SNPs in significant association with longevity that gave a *p*-value of 0.001 (−log10 value of 3). In addition, highly significant *p*-values were calculated by taking the reciprocal of the number of markers [13]. Thus, SNPs with *p*-values of 4.26 × 10^−5^ were considered to be highly significantly associated.

### 4.5. Blast Analysis

In order to look for the candidate genes linked with the associated SNPs, sequence retrieval of each significant or highly significant SNP was performed. This was achieved by retrieving a raw sequence of 301 nucleotides considering 150 bases upstream and 150 bases downstream of each candidate SNP. The sequence was generated from the reference genome *C. annuum* CM334 [68] version 1.6 using samtools faidx [69] via a blast analysis of all the retrieved sequences of associated SNPs. These sequences were used as a query in the NCBI BLASTX (https://blast.ncbi.nlm.nih.gov/Blast.cgi?PROGRAM=blastx&PAGE_TYPE=BlastSearch&LINK_LOC=blasthome, accessed on 3 January 2023) research tool database for functional gene annotations. The topmost hits with the smallest *E*-value and a high percentage of query coverage were reported as potential candidate genes.

## 5. Conclusions

To conclude, we presented the very first comprehensive genetic analyses of seed longevity in *Capsicum* using the real-time data after long-term cold storage and the untapped natural genetic diversity. Several candidate genes have been reported for seed longevity in *Capsicum.* Some of them have already been reported for longevity in wheat, barley or other crops, whereas others are novel. Our findings lay the foundation for the comprehensive future studies of seed longevity in *Capsicum*.

## Figures and Tables

**Figure 1 plants-12-01321-f001:**
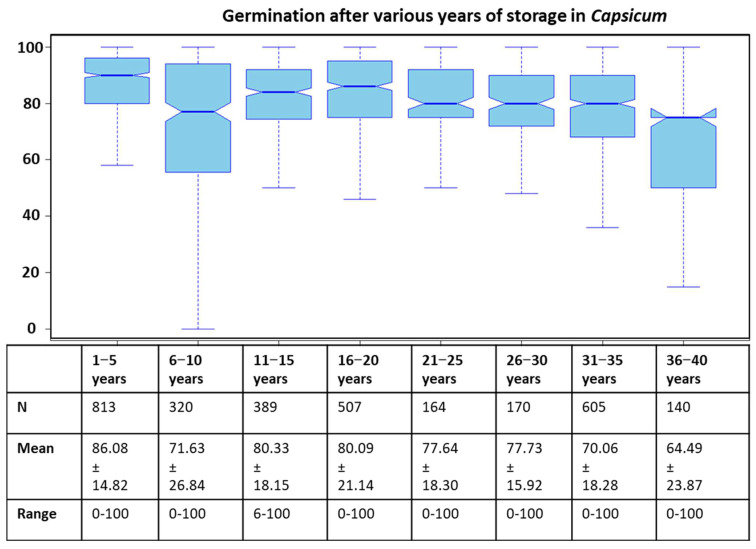
Mean (±standard deviation) germination percentages (seed survival) after various years of storage in *Capsicum*, where N = sample size.

**Figure 2 plants-12-01321-f002:**
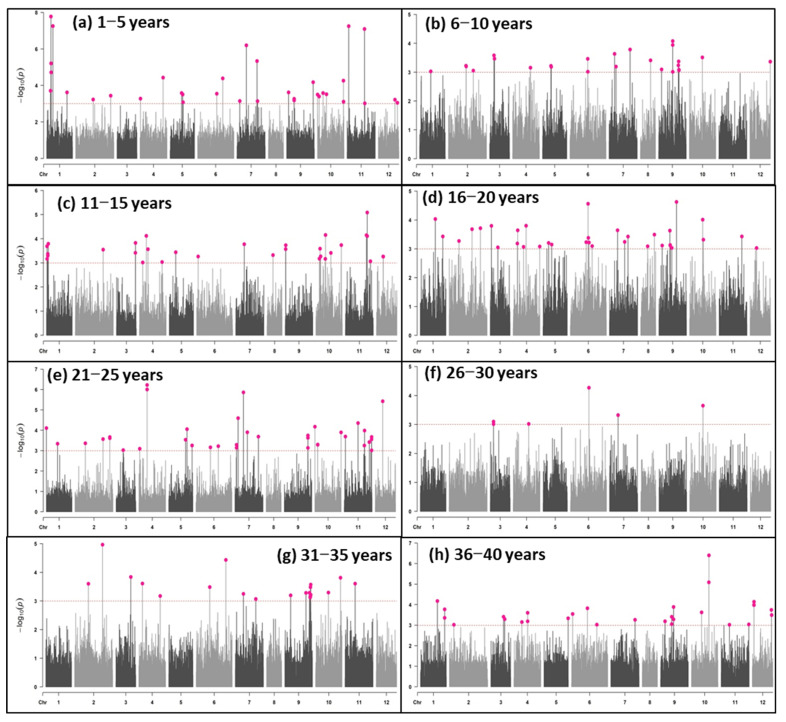
The plot of a genome-wide scan (GWA analysis) of SNP markers associated with seed longevity over various periods of storage ((**a**) after 1–5 years of storage, (**b**) after 6–10 years of storage, (**c**) after 11–15 years of storage, (**d**) after 16–20 years of storage, (**e**) after 21–25 years of storage, (**f**) after 26–30 years of storage, (**g**) after 31–35 years of storage and (**h**) after 6–10 years of storage) in *Capsicum* accessions. The chromosomes are shown on the *x*-axis, the genome-wide scan −log10 (*p* values) is shown on the *y*-axis, and the significantly associated SNPs are highlighted in pink.

**Figure 3 plants-12-01321-f003:**
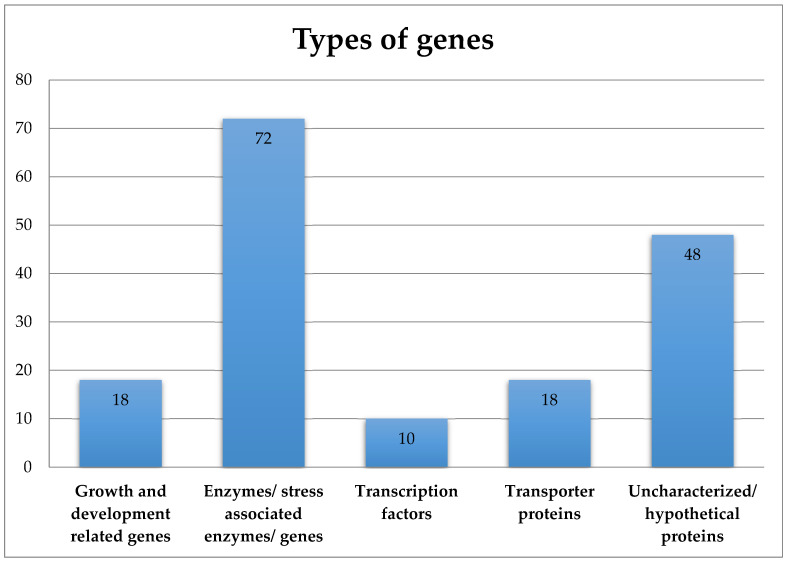
Types of candidate genes linked with the longevity-associated SNPs.

**Figure 4 plants-12-01321-f004:**
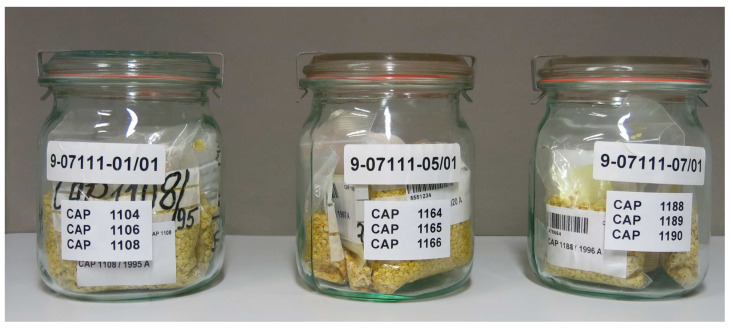
Storage of various *Capsicum* accessions [CAP1104, CAP1106, CAP1108, CAP1164, CAP1165, CAP1168, CAP1188, CAP1189 and CAP1190 (for details see Appendix A) using the genebank ID number] in glass containers kept at below-freezing temperatures in the IPK genebank.

## Data Availability

Not applicable.

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
