# Peer review of "Genetic Analyses of Seed Longevity in *Capsicum annuum* L. in Cold Storage Conditions"

_plants, 2023, doi:10.3390/plants12061321_

Round 1
Reviewer 1 Report
Storage conditions, especially long-term, as well as the quality of seed, genetic and breeding material are the most important components of breeding work, which makes the article relevant. The authors may consider various storage options, with the suggestion of the most optimal mode, to increase germination taking into account the data obtained.
The selection of material for selection is the most important stage on which efficiency directly depends, as shown in the article (Shapturenko et al., 2014) "Dna divergence as a criterion of a sweet pepper (Capsicum annuum L.) selection for heterosis". With the development of genetic research methods, genetic markers of selective breeding traits for productivity and resistance to stress, pathogens and pests are evaluated (Porotnikov et al., 2020; Saleem et al., 2022; Akram et al, 2022; Sharma et al, 2022). On the other hand, biochemical markers of resistance are also used (Li et al., 2010), characteristic of natural halophyte populations (Rozentsvet., Nesterov, 2021), as well as cultures (Rozentsvet et al, 2021; Bakunov et al, 2022). The authors are recommended: 1. consider various storage options, with the suggestion of the most optimal mode, to increase germination taking into account the data obtained; 2. note the presence of genetic markers; 3. describe in detail the conditions of seed germination.
Author Response
Reviewer 1
Many thanks for your time and constructive criticism. We are providing response to each query below:
Consider various storage options with the suggestion of the most optimal mode in increase germination taking into account the data obtained.
Response: It is mentioned in our MS that the storage conditions for the stored seeds were -15 °C till the year 2011. Afterwards, the protocol was revised and the new storage conditions were -18 °C for the rest of the storage period. It is pertinent to mention here that the federal genebank at the IPK tries its level best to maintain the above mentioned storage conditions throughout the storage period of a given seed lot. The genebank is also ISO 9001:2015 certified. In addition, these conditions are approved by International Seed Testing Association which is adjusted from time to time depending on new research or findings.
Hence, the storage conditions where our seeds were stored were the best possible practical storage conditions. However, what was strikingly different was the germination behavior over the period of storage. The main task of this investigation was to know what causes that differential behavior at the genetic level which we have presented very thoroughly.
Note the presence of genetic markers
Response: There are numerous kinds of molecular markers now available for almost all the crops. In particular, with the outsourcing of the genotyping, much more maps are now available and different types of molecular markers are now under investigation. With regards to Capsicum, we used the genotyping-by-sequencing SNPs. For complexity reduction, a two-enzyme protocol using PstI (CTGCAG) and MspI (CCGG) was used and sequencing was carried out using an Illumina HiSeq2500 platform generating 1×107-bp single-end reads version 3 chemistry (Illumina). Finally, the trimmed reads from sequencing were then aligned to reference genome sequence C. annuum CM334 version 1.6 available at http://peppergenome.snu.ac.kr using BWA-MEM version 0.7 (40) and converted to binary alignment map format using SAMtools.
In our tables (Table S1 and S3 and S4), the markers (SNPs) are represented by chromosome number viz. S1 (SNP on chromosome 1 of Capsicum) followed by underscore (_) and the base pair. Thus, it is clearly evident from the very first marker provided in Table S1 (SNP: S1_133318) that this SNP is on chromosome 1 at the 133318 base pair position.
Describe in detail the conditions of seed germination
Response: The seeds were germinated on round filter paper with glass covers on Jacobsen Apparatus at 25 ± 2°C and 23 ± 2°C during the day and night, respectively. The germination percentages were recorded on the eighth day according to International Seed Testing Association (ISTA) protocols (i-e when enough of the roots and shoots have emerged that could survive). An example of how seeds would appear on the filer paper after germination is provided on the picture below (Fig 1):
Fig 1 Germinated seeds of Capsicum on filer paper on the eighth day of the experiment

Reviewer 2 Report
The manuscript Genetic analyses of seed longevity in Capsicum annuum L. in cold storage conditions by Mian Abdur Rehman Arif et al presents an interesting study related to determined seed longevity markers in C. annum.
The introduction is well written and point out the relevance of using these genetic analyses to study seed longevity.
The results related to seed germination are not clearly presented. There are no statistical differences between groups? You mention that direct comparison among the results of different intervals is not possible because of the involvement of dissimilar accessions tested during each interval but there are no statistical differences between 1-5 years and 36 to 40 years groups?
Considering the results, it will be possible that you also presented the different MTAs in functional groups in the discussion section to better understand their physiological relevance. If there is an statistical difference between the 1-5 years and 36 to 40 years groups, it will be possible to compare MTAs between them?
There are many genes that are not in cursives.
Author Response
Reviewer 2
Many thanks for your time and constructive criticism. We are providing response to each query below:
The manuscript Genetic analyses of seed longevity in Capsicum annuum L. in cold storage conditions by Mian Abdur Rehman Arif et al presents an interesting study related to determined seed longevity markers in C. annum.
The introduction is well written and point out the relevance of using these genetic analyses to study seed longevity.
The results related to seed germination are not clearly presented. There are no statistical differences between groups? You mention that direct comparison among the results of different intervals is not possible because of the involvement of dissimilar accessions tested during each interval but there are no statistical differences between 1-5 years and 36 to 40 years groups?
Considering the results, it will be possible that you also presented the different MTAs in functional groups in the discussion section to better understand their physiological relevance. If there is an statistical difference between the 1-5 years and 36 to 40 years groups, it will be possible to compare MTAs between them?
There are many genes that are not in cursives.
- The results related to seed germination are not clearly presented. There are no statistical differences between groups? You mention that direct comparison among the results of different intervals is not possible because of the involvement of dissimilar accessions tested during each interval butthere are no statistical differences between 1-5 years and 36 to 40 years groups?
Response: It is indeed true that no direct comparison is possible. However, we did state that, “there were 171, 261, 419, 163, 108, 318 and 54 accessions from 1-5 year interval that were common with 6-10 years, 11-15 years, 16-20 years, 21-25 years, 26-30 years, 31-35 years and 36-40 years intervals, respectively. ANOVA results indicated that germination % among common accessions was significantly different in all such cases”. Since we submitted our MS as communication, we did not present the ANOVA results as a separate section in the running MS. However, for your reference, the results of the single factor ANOVA of the similar accessions stored for 1-5 years and up to 10 years and up to 15 years are being presented.
Single factor ANOVA between the common accessions of 1-5 and 6-10 years storage interval
|
SUMMARY |
||||||
|
Groups |
Count |
Sum |
Average |
Variance |
||
|
G5 |
171 |
14570 |
85.20468 |
196.0579 |
||
|
G10 |
171 |
12398 |
72.50292 |
846.0162 |
||
|
ANOVA |
||||||
|
Source of Variation |
SS |
df |
MS |
F |
P-value |
F crit |
|
Between Groups |
13794.11 |
1 |
13794.11 |
26.47433 |
4.5237E-07 |
3.868954199 |
|
Within Groups |
177152.6 |
340 |
521.037 |
|||
|
Total |
190946.7 |
341 |
|
|
|
|
Single factor ANOVA between the common accessions of 1-5 years and 11-15 years storage interval
|
Anova: Single Factor |
||||||
|
SUMMARY |
||||||
|
Groups |
Count |
Sum |
Average |
Variance |
||
|
Gr5 |
261 |
22464 |
86.06897 |
230.4106 |
||
|
Gr15 |
261 |
20838 |
79.83908 |
383.974 |
||
|
ANOVA |
||||||
|
Source of Variation |
SS |
df |
MS |
F |
P-value |
F crit |
|
Between Groups |
5064.897 |
1 |
5064.897 |
16.48771 |
5.6510E-05 |
3.859404 |
|
Within Groups |
159740 |
520 |
307.1923 |
|||
|
Total |
164804.9 |
521 |
|
|
|
|
Likewise, the results of the single factor ANOVA of the similar accessions stored for 1-5 years and up to 35 years and up to 40 years are being presented. Since the p-value is < 0.001, and our F-value is greater than critical value of F, the germination percentage (survival) was significantly different among the groups.
Single factor ANOVA between the common accessions of 1-5 years and 31-35 years storage interval
|
SUMMARY |
||||||
|
Groups |
Count |
Sum |
Average |
Variance |
||
|
Gr5 |
318 |
27582 |
86.73585 |
245.7407 |
||
|
Gr35 |
318 |
22879 |
71.94654 |
410.2653 |
||
|
ANOVA |
||||||
|
Source of Variation |
SS |
df |
MS |
F |
P-value |
F crit |
|
Between Groups |
34777.06 |
1 |
34777.06 |
106.0266 |
4.26E-23 |
3.856168 |
|
Within Groups |
207953.9 |
634 |
328.003 |
|||
|
Total |
242731 |
635 |
|
|
|
|
Single factor ANOVA between the common accessions of 1-5 years and 36-40 years storage interval
|
SUMMARY |
||||||
|
Groups |
Count |
Sum |
Average |
Variance |
||
|
Gr5 |
55 |
5077 |
92.30909 |
142.2175 |
||
|
Gr40 |
55 |
2819 |
51.25455 |
653.6007 |
||
|
ANOVA |
||||||
|
Source of Variation |
SS |
df |
MS |
F |
P-value |
F crit |
|
Between Groups |
46350.58 |
1 |
46350.58 |
116.4854 |
7.31E-19 |
3.929012 |
|
Within Groups |
42974.18 |
108 |
397.9091 |
|||
|
Total |
89324.76 |
109 |
|
|
|
|
- Considering the results, it will be possible that you also presented the different MTAs in functional groups in the discussion section to better understand their physiological relevance
Response: Based on your suggestion and our understanding we have tried to group the identified genes based on their function. For example, we have stated that “blast analyses of the reported SNPs identified several candidate genes for longevity (Table S3). Of the 220 associated SNPs, 167 SNPs successfully provided hits with certain candidate genes. These 167 SNPs could further be divided further into five groups based on the function they perform (Fig 3, Table S3). The first group included ten genes which were mainly involved in growth and development related processes. The other group constituted 72 genes which were mainly enzymes either specifically produced under (both biotic and abiotic) stress or under normal conditions. The third group constituted 10 genes that were mainly transcription factors. The fourth group included 18 genes that were mainly transporter genes whereas the fifth group constituted 48 genes that were mainly uncharacterized and/or hypothetical proteins”. The text in blue has been incorporated in the running text
Fig 3 Types of candidate genes linked with the longevity associated SNPs
- If there is an statistical difference between the 1-5 years and 36 to 40 years groups, it will be possible to compare MTAs between them?
Response: We did try to compare the MTAs between the above mentioned two groups and we could not find any similarity between the two groups of MTAs.
We further tried to look at the candidate genes of the associated SNPs of the two groups based on function. We had 29 and 23 genes identified for 1-5 years and 36-40 years of storage, respectively. In the below figure, the percentage of genes identified belonging to different groups based on their function is presented in percentages. One important feature that we detected was that the hypothetical proteins were doubled in case of 36-40 years of storages (~48%). One reason for that could be that such proteins have not been investigated in detail because the plant material that carries these proteins is rare. Hence, we conclude that the germplasm used in this investigation is really very unique and provides the basis for subsequent planned studies on longevity.
“Since we are presenting our findings as “communication” we did not include that part in the discussion”.
- There are many genes that are not in cursives.
Response: We have made sure that all genes are presented in italic

Round 2
Reviewer 2 Report
The authors have answered all my questions. I consider that the manuscript is ready to be published.
Author Response
Dear Editor,
Many thanks for sparing the time for this manuscript. According to the reviewer 2, we have answered all the questions and the MS is ready to be published.
On a side note we have also answered all the queries from reviewer 1 and there are no further comments from that reviewer. Hence, we assume that both the esteemed reviewers (reviewer 1 and reviewer 2) are satisfied and the MS does not need any further refinement at the moment.
On the behalf of all the co-authors,
Mian Abdur Rehman Arif